# The Effect of the Averaging Period for PMF Analysis of Aerosol Mass Spectrometer Measurements during Off-Line Applications

Christina Vasilakopoulou[1,2], Iasonas Stavroulas[3,4], Nikolaos Mihalopoulos[3,4] and Spyros N. Pandis[1,2]

[1]Department of Chemical Engineering, University of Patras, Patras, Greece
[2]Institute of Chemical Engineering Sciences, ICE-HT, Patras, Greece
[3]Department of Chemistry, University of Crete, Heraklion Crete, Greece
[4]Institute for Environmental Research and Sustainable Development, National Observatory of Athens, Athens, Greece

*Correspondence to*: Spyros N. Pandis (spyros@chemeng.upatras.gr)

**Abstract.** Off-line Aerosol Mass Spectrometer (AMS) measurements can provide valuable information about the ambient organic aerosol in areas and periods in which online AMS measurements are not available. However, these offline measurements have low temporal resolution as they are based on filter samples collected usually over 24 hours. In this study we examine whether and how this low time resolution affects source apportionment results. We used a five-month period (November 2016-March 2017) of online measurements in Athens and performed Positive Matrix Factorization (PMF) analysis to both the original dataset, which consists of 30 min measurements, and to time averages from 1 up to 24 h. The 30 min results indicated that five factors were able to represent the ambient organic aerosol (OA): a biomass burning organic aerosol factor (BBOA) contributing 16% of the total OA, hydrocarbon-like OA (HOA) (29%), cooking OA (COA) (20%), more oxygenated OA (MO-OOA) (18%), and less oxygenated OA (LO-OOA) (17%). Use of the daily averages resulted in estimated average contributions that were within 8% of the total OA compared with the high-resolution analysis for the five-month period. The most important difference was for the BBOA contribution which was overestimated (25% for low resolution versus 17% for high resolution) when daily averages were used. The estimated secondary OA varied from 35 to 28% when the averaging interval varied between 30 min and 24 h. The high-resolution results are expected to be more accurate, both because they are based on much larger datasets and because they are based on additional information about the temporal source variability. The error for the low-resolution analysis was much higher for individual days and its results especially for high concentration days are quite

uncertain. The low-resolution analysis introduces errors in the determined AMS profiles for the BBOA and LO-OOA factors but determines the rest relatively accurately (theta angle around 10$^{\circ}$ or less).

## 1 Introduction

Exposure to high concentrations of particulate matter (PM) can lead to major health problems including stroke, heart disease and lung cancer (WHO, 2004; Pope and Dockery, 2006). Around 90% of the world population lives in places where air pollution exceeds the World Health Organization limits and more than 4 million people die every year due to ambient air pollution (WHO, 2018). To decrease the levels of PM it is necessary to identify its sources in each area and quantify their contributions.

Receptor models have been used for decades for the quantification of atmospheric aerosol sources (Hopke, 1991). Positive Matrix Factorization (PMF) (Paatero and Tapper, 1994) is the most widely used approach for the organic aerosol (OA) Aerosol Mass Spectrometer measurements. PMF constraints result in non-negative solutions, which make it suitable for the analysis of environmental data. PMF has been used in many studies (Ulbrich et al., 2009; Aiken et al., 2009; Docherty et al., 2011) in order to estimate the sources of the OA and it does not require a-priori information about the profiles. However, there are cases in which PMF can result in mixed or non-meaningful factors (Canonaco et al., 2021). In these cases the multilinear engine algorithm (ME-2) (Paatero, 1999; Canonaco et al., 2013) can be used. ME-2 has the advantage that pre-determined factors can be assumed by the user and, with a certain degree of freedom, will be part of the final solution (Crippa et al., 2014).

One of the challenges of PMF application on OA AMS spectra is that the factor profiles may change over time. This is especially true for oxygenated OA (OOA) factors (Freney et al., 2011; Dai et al., 2019; Via et al., 2021). When periods with different aerosol chemistry are mixed, such as summer with winter months, important information can be lost (Xie et al., 2013; Canonaco et al., 2015; Reyes-Villegas et al., 2016). For that reason many studies that examine long-term datasets break them up to monthly or seasonal subsets (Xu et al., 2015; Budisulistiorini et al., 2016; Hu et al., 2017)

that are analyzed separately. The use of a rolling window has been proposed for the analysis of large datasets (Parworth et al., 2015; Chen et al., 2020; Canonaco et al., 2021) thus avoiding choosing periods by trial and error.

The robustness of the PMF results depends on the number of samples used; the relative errors increase as the sample dimension decreases (Hedberg et al., 2005). Typically at least 60-200 sets of observations are used for PMF analysis of individual aerosol components (Jaeckels et al., 2007). Hedberg et al. (2005) compared the results derived from 80 $PM_{10}$ measurements of 26 elements and from randomly reduced subsets, containing 85%, 70%, 50% and 33% of the initial samples. A five-factor solution was determined in each case. The results of the analysis of the reduced subsets showed that the source contribution of each factor to the total OA differed by less than 10% with the contributions derived from the initial dataset. Zhang et al. (2009) analyzed a total of 273 samples for 46 VOCs, organic and elemental carbon and silicon. Multiple subsets, including approximately half of the observations (135 samples), 33% (90 samples) and 20% (54 samples) were also analyzed. The results of the 50 samples datasets had high relative standard deviations (above 50%) for the contribution of some factors with low OC concentrations. These suggest that the corresponding results were quite uncertain. There have been studies that attempted PMF analysis for PM elements using only 30-50 samples (Sunder Raman and Ramachandran, 2010; Tiwari et al., 2013; Manousakas et al., 2015). For example Manousakas et al. (2017) used a dataset of 55 samples with 22 elements and identified 6 factors. The solution was relative robust suggesting that the sample size was sufficient for the purposes of these studies.

Other studies have examined the source apportionment of different time resolution inputs of VOCs, metals, or combinations of inorganic ions with metals or VOCs with metals. Peng et al. (2016) measured 15 $PM_{2.5}$ metals, organic and elemental carbon and 6 inorganic ions with 1 h time resolution in Beijing, resulting in 528 samples. ME-2 was conducted at four temporal resolution settings (1, 2, 4 and 8 h) and a four-factor solution was obtained in each case. The biggest discrepancy among the contributions of each factor to the total $PM_{2.5}$ was observed for coal combustion which varied from 15% of the total $PM_{2.5}$ (for the 1 h) to 29% (for the 4 h); in the 8 h analysis it was 27%. Wang et al. (2018) examined the impact of time resolution on PMF results, by averaging the initial 512 1 h

resolution samples of 20 PM components (13 elements, 4 inorganic components, OC, EC and $PM_{2.5}$ mass) to 4 h (145 samples) and 6 h (97 samples) time intervals. Even though the same eight factors were identified for every averaging interval, 3 of them showed large variation in the average contribution in the low resolution cases. Yu et al. (2019) used 1 h (N=6456) measurements of 16 metals and averaged them over 23 h (N=297). The 23 h PMF analysis overestimated the mass concentration for 2 out of the 6 factors but gave consistent factor contributions with the 1 h solution.

Off-line AMS analysis was introduced by Daellenbach et al. (2016). Filter samples ($PM_1$, $PM_{2.5}$ and $PM_{10}$) are extracted in ultrapure water. The water extracts are filtered and aerosolized, thus converted to droplets which are dried and measured with an AMS. The 24 h average AMS spectra from the offline and the online analysis were highly correlated ($R^2$>0.97) (Daellenbach et al., 2016). The PMF results obtained from 24 h filter samples were very different from collocated on-line measurements, even though there were not such differences in the input spectra. It is not clear if discrepancies could be due to the temporal resolution of the analysis or due to experimental issues such as the blank uncertainty, the sample extraction efficiency, potential filter sampling artifacts, etc.

Even though there are studies which have examined the effect of time resolution for several chemical species such as metals and VOCs, it is not yet clear whether the low temporal resolution in the off-line AMS analysis introduces significant errors in the estimated contributions of different sources. To explore this, we conducted PMF analysis for an ambient OA dataset averaged in different resolutions (from 1 h up to 24 h) and compared the results with the initial resolution of the dataset which was 30 min. Our objective in this work is to quantify the effect of the use of low-temporal resolution data in the PMF analysis of AMS measurements without necessarily considering the high temporal resolution results as the "truth". Clearly, even at the highest temporal resolution the OA source apportionment using AMS measurements has uncertainties and errors as any source apportionment technique. The longer-term aim of this study is to characterize step by step the uncertainty of the PMF analysis of off-line AMS applications, neglecting at this stage the uncertainty arising from the various sampling and extraction artifacts. These experimental issues can also lead to differences between the results of PMF when applied to on-line AMS measurements and to the off-

line AMS measurements of the extracts of daily filter samples. Nevertheless, both the on-line and off-line techniques can clearly provide useful information about the OA sources.

## 2 Measurements and their PMF analysis

In this study, a dataset obtained by an Aerosol Chemical Speciation Monitor (ACSM) (Aerodyne Inc.,
USA) is used, operated at the National Observatory of Athens (NOA) at Thissio, in the center of Athens. The measurement resolution was 30 min. Measurements lasted one year, beginning in July 2016 and ending in August 2017.

For the PMF analysis the SoFi (Source Finder) version 6.1 graphic interface (Canonaco et al., 2013) was used. OA unit mass resolution spectra ($m/z$ 12-125) were analyzed. The error matrix was
120 weighted using a step function, as proposed by Paatero and Hopke (2003), and a cell-wise signal-to-noise ratio (S/N) was calculated (Brown et al., 2015). "Bad" signals with S/N below 0.2 were down-weighted by a factor of 10, "weak" signals with S/N between 0.2 and 1 were down-weighted by a factor of 2 and the $CO_2$ related variables ($m/z$ 16, 17, 18, 44) were also down-weighted by a factor of 2 (Ulbrich et al., 2009). The minimum *Fpeak* value was -1 and the maximum 1. The *Fpeak* step was
125 0.1. The optimum *Fpeak* was chosen each time based on the physical meaning of the factors, the resulting spectral profiles and their diurnal variation of the factor levels. The factor profiles were compared with those in the literature and their average diurnal variation was compared with the results of previous studies in Greece during similar cold periods.

Given that the high temporal resolution dataset has a 30 min temporal resolution, the relatively
high concentration data points ("spikes") were kept in the dataset to avoid loss of information about the sources of primary organic aerosol. To explore the effect of these periods on our results, we have analyzed seven days with the highest observed 30 min OA concentrations (all above 100 μg m$^{-3}$) during the five-month period examined in this work (Fig. S10). These high OA concentrations were observed at nights, between 21:00 and 3:00 and remained high for several hours (Fig. S11). The results
of the comparison of the 24 h and 30 min PMF results during these days with high concentration periods were quite consistent with the rest of the days for the primary OA components (Fig. S12).

Overall, we did not observe a notable change in the behavior of the PMF analysis using low temporal resolution data during these interesting high concentration events.

## 3 Results and discussion

### 3.1 Monthly PMF analysis of the full annual dataset

The full one-year dataset was initially analyzed by month at the highest time resolution, which was 30 min. The contribution of each factor to the total OA for each month is shown in Fig. 1. The PMF analysis for the summer months (June, July and August) showed that four factors could represent the ambient OA: two primary (COA and HOA) and two secondary (MO-OOA and LO-OOA). Four factors were also identified for the first two autumn and the last two spring months. The PMF analysis for December, January and February showed the presence of an extra BBOA factor. The same additional factor was found for November and March. These results are consistent with those of Stavroulas et al. (2019) where PMF analysis was done separately for a "cold period" (November to March) and a "warm period" (May to September). Again, four factors (HOA, COA, MO-OOA, and LO-OOA) were found for the "warm period", while in the "cold period" an additional BBOA factor was identified.

The subset that was used in this study was the five-month cold period, beginning at November 1 2016 and ending at March 18, 2017, resulting in a set of 6150 30-min samples. This period was chosen, because of the presence of the BBOA factor in the PMF analysis, which is an additional primary factor. The average organic mass concentration for this cold period was 9.2 µg m$^{-3}$ with a maximum concentration of 201 µg m$^{-3}$ (Fig. 2). From now on all the results will refer to this cold time period.

### 3.2 High temporal resolution PMF analysis

The 6150 30-min measurements during the cold period were analyzed together. Five factors could represent the variation of the organic aerosol ACSM spectra based both on the residuals and the physical meaning of the solutions (Figs. S1-S3). Three of them were primary (HOA, COA and BBOA) contributing 65% to the total OA, and two were secondary (MO-OOA and LO-OOA) with a

contribution equal to 35%. The same result for all practical purposes was observed from the analysis of the measurements during each month separately, with an average primary contribution of 63%. We will focus first on the analysis of the full dataset (all months together) and then discuss the analysis of the data of each month separately.

HOA was the biggest contributor (29%) to the total OA for the cold period. The average HOA concentration peaked at 9:00 local time (LT) (Fig. 3), which is consistent with the local rush hour. Its mass spectrum was characterized by $m/z$'s 41, 43, 55 and 57 (Fig. S4) (Mohr et al., 2009). COA was the second biggest contributor to the total OA, representing 20% of the total OA. The COA mass spectrum had a strong peak at $m/z$ 41, and a high ratio of $m/z$ 55/57. This high ratio characterizes COA emissions in urban areas (Sun et al., 2011). The average COA concentration increased during the late afternoon and night hours (Fig. 3). This is consistent with the activity patterns of the restaurants in Athens during this colder period of the year. BBOA represented 16% of the total OA. Its maximum 30 min concentration reached 58 µg m$^{-3}$ (Fig. S5). The distinguishing feature of BBOA is the presence of strong signals at $m/z$'s 60 and 73 (Alfarra et al., 2007, Ng et al., 2011b). The diurnal profile of the BBOA showed an increase at 18:00 LT reaching a peak at 23:00 LT. This 18:00-23:00 LT period is consistent with the times that fireplaces are used in Athens, a common indoor heating process in Greece during the last decade. MO-OOA, representing 18% of the total OA, had little average diurnal variation, and an average concentration of 1.7 µg m$^{-3}$. On the other hand, LO-OOA (17% of the total OA) increased during night-time when primary OA also increased. This is consistent with local night time production during wintertime (Kodros et al., 2020). The two secondary factors were separated due to their differences in specific $m/z$ values like 43 and 44. The MO-OOA mass spectrum had a strong peak at $m/z$ 44, while the LO-OOA mass spectrum was characterized by a strong signal at $m/z$ 43, and a lower one at $m/z$ 44. The PMF results from this study agreed (within 20%) with the Stavroulas et al. (2019) unconstrained PMF analysis. Stavroulas et al. (2019) used as inputs in their analysis factor profiles for the BBOA, COA and HOA allowing ME-2 a certain degree of freedom around these inputs. In the present study the unconstrained solution is used in order to avoid the additional complexity that the use of external factors may introduce. Detailed comparisons between the unconstrained solutions of the studies can be found in the SI (Figs. S6 and S7).

### 3.3 Comparisons between PMF results at different temporal resolution

The goal of this study is to examine whether the PMF results change as the sampling time resolution decreases. For this reason, we calculated the 1, 2, 4, 6, 8, 10, 12 and 24 h averages of the measured OA spectra and performed PMF analysis in each new dataset. We do not consider the high temporal resolution results as the "truth" because clearly, they have their own errors characteristic of any source apportionment technique. The number of samples used in each averaged dataset were above 100 in each case (Table S1). To avoid unnecessary complications, the uncertainty was simply averaged for the different temporal resolution datasets in the main analysis. Five factors were able to explain the OA variation in all cases. The estimated primary factor (HOA, COA and BBOA) contribution to the total OA ranged from 72% (at 24 h) to 58% (at 10 h resolution). The 30 min resolution analysis suggested that 65% of the total OA was primary (Fig. 4).

The HOA contribution to the total OA ranged from 29% (30 min) to 23% (daily resolution) (Fig. 5). The 30 min COA contribution was 20% of the OA. The minimum COA observed was 19% (4 h) and the maximum 25% (daily resolution). The BBOA contribution varied the most among the primary factors and ranged from 15% (1 h) to 24% (daily resolution). In the 30 min solution the BBOA contribution was 16% of the OA.

In the 30 min analysis the estimated LO-OOA contribution to the total OA was 17%. The LO-OOA showed a quite sensitive and unstable behavior, as it ranged from 10% (6 h) to 21% (2 h), and for the daily resolution was 13%. The 30 min MO-OOA was 18% of the OA and ranged from 15% (daily resolution) to 24% (6 h).

The variation of the spectra of the various factors resulting from the analysis at different averaging periods was quantified using the theta angle (Kostenidou et al., 2009). In this approach the spectra are treated as vectors and theta is the angle between them. A theta angle below 15° indicates that the two factors are quite similar. The highest angle calculated between the different resolution spectra with the 30 min ones for the COA was 26° (for the 6 h resolution). This corresponds to an $R^2$ equal to 0.76. For HOA the highest angle was 19° (for the 10 h resolution, $R^2 = 0.87$) and for BBOA 22° (daily resolution, $R^2 = 0.80$) (Fig. 6). This indicates that in these cases the mass spectra were quite different from the 30 min spectrum.

The BBOA mass spectrum from the analysis of the 24 h analysis is more similar with BBOA spectra in the literature compared to that from the analysis of the 30 min data. One of the reasons is that the BBOA spectra in the literature are mostly based on the analysis of AMS and not ACSM results. AMS and ACSM spectra for the same factor can be different because of the different fragmentation tables used in the analysis of the measurements of the two instruments. The theta angle between the 2 h and the 30 min BBOA is equal to 19° which shows that the two spectra have some significant differences mainly in $m/z$ 18, 41 and 55 (Fig. S8). On the other hand, the signal at $m/z$ values 60 and 73, which are characteristic of BBOA, was in good agreement between the two temporal resolution results. In the case of the 4 h averages the theta angle of the BBOA factor compared with the 30 min BBOA is 14°, which indicates that the two factors are relatively similar. Additional information about the spectral comparisons can be found in the SI (Figs. S8-S9).

The MO-OOA spectrum remained relatively similar to the 30 min one for all temporal resolutions, with the highest angle being 11° (for daily resolution, $R^2 = 0.97$). On the other hand, the LO-OOA spectrum was the most variable varying by as much as 30° ($R^2 = 0.68$) compared to the high temporal resolution spectrum. The two secondary factors were separated due to their differences in specific $m/z$ values, like 43 and 44, but also due to their different atomic oxygen to carbon (O:C) ratios. At high temporal resolution, the two factors can be better separated from each other by PMF. On the contrary, for the low temporal resolution data, mixing of the two secondary factors is observed. The MO-OOA factor location (Fig. 7) in the $f_{44}$ vs $f_{43}$ plot (Ng et al., 2011a) tended to move down approaching the LO-OOA factor as the temporal resolution was decreased. On the other hand, LO-OOA was moving upwards in the plot. The MO-OOA O:C decreased from 1.09 to 0.88 as the time resolution decreased. The LO-OOA O:C increased from 0.32 for the 30 min resolution to 0.6 for the daily resolution (Fig. 8). This change in the LO-OOA spectrum and contribution appears to be one of the major effects of the PMF analysis time resolution.

The most important reason for the observed discrepancies is probably the reduction of information provided to PMF when one moves from thousands of measurements (for the 30 min dataset) to a little more than one hundred (for the 24 h resolution data). Given that the diurnal variation of the source contributions is lost during this averaging, it is quite surprising that the differences that

we found are that low. Higher discrepancies were observed in the spectra than in the factors contribution, comparing the low and high temporal resolution results. However, the changes in the spectra between the different temporal resolution results were due to $m/z$ values which were not that important for the identification of each factor. All the specific source-specific markers appeared in the PMF solution at every averaging interval, making the identification and quantification of each factor possible. The comparison of the source spectra derived from the low and high temporal resolution PMF analysis, is depicted in Fig. 9. One should note that the low resolution ACSM mass spectra used in the present work probably represent a worst-case scenario for the uncertainty of the off-line AMS analysis in general. One would expect that the use of high-resolution AMS spectra will result in even lower uncertainty. The magnitude of this potential reduction of this uncertainty moving to high resolution off-line AMS analysis is a topic for future investigation.

Our analysis so far has focused on unconstrained PMF solutions in order to avoid the additional complexity introduced by the use of external factors which may or may not be representative of the OA sources in the corresponding area. We repeated the PMF analysis constraining the primary factors (HOA, BBOA and COA) for both the 30 min and the 24 h resolution. The profiles suggested by Ng et al. (2011b) were used to constrain the HOA (a=0.1) and BBOA factors (a=0.4) and the profile of Crippa et al. (2013) for COA (with a=0.2) (Figs. S13-S17). The results of the constrained analysis at the two temporal resolutions were quite consistent (discrepancies less than 15%) with each other (Fig. S18). Details can be found in section 6 of the Supplementary Information. Therefore, our conclusions about the role of the temporal resolution are quite robust also in this case of constrained analysis.

We have also performed a sensitivity test using a geometric average for the calculation of the error matrix of the 24 h results. The arithmetic average was used for the AMS measurements. In this rather extreme sensitivity test, the predicted contribution of each primary factor to the total OA changed by less than 10% compared with the PMF results using the arithmetic average error (Fig. S20). The highest discrepancy was observed for MO-OOA and was 16%. Once more the estimated AMS spectra from the PMF showed higher discrepancies than the source contributions (Fig. S21).

The theta angle of the COA spectra using the geometric and the arithmetic average error was 30°. On the other hand, the best agreement was observed between the two MO-OOA spectra (8°).

## 3.4 Analysis of the 24 h resolution results for each day

The daily resolution was the lowest resolution used in this work and is the usual resolution for off-line AMS analysis. The number of samples in this case was 127. In the 24 h solution the primary factors represented 72% of the total OA compared to 65% for the 30 min. Considering the uncertainty of the source apportionment approaches like PMF, a 7% change of a source contribution is of secondary importance and is within the error margin of the analysis. So, the use of the daily resolution measurements does not introduce significant errors in the primary/secondary OA split of the AMS analysis on average. While the ability of the low temporal resolution results to determine average contributions during the study period is encouraging, it is interesting to examine its performance for individual days. We estimated the daily average concentrations of the concentrations of the five factors from the 30 min analysis and we compared them with the results of the 24 h analysis for each day (Fig. 10).

For HOA the results of the two approaches were in encouraging agreement during most of the days and were well correlated ($R^2$=0.96). The tendency of the 24 h analysis to underestimate the HOA was evident during most days. During some days with relatively low HOA levels (below 2 µg m$^{-3}$) there were significant errors with the 24 h analysis estimating practically zero HOA and seriously underestimating its levels. Despite these discrepancies a relatively good consistency in the estimates of the two approaches for the HOA role is observed, as the 30 min solution HOA (29% of the total OA) agrees well with that estimated from the 24 h resolution (23%).

The behaviour of the low temporal resolution PMF analysis for the COA was the opposite of that for the HOA. The COA was systematically overestimated during the high COA concentration (above 4 µg m$^{-3}$) periods (Fig. 10). In these days the 24 h COA resolution results were as much as two times higher than the 30 min results. For example, the highest COA concentration estimated by the high-resolution analysis was 5.7 µg m$^{-3}$ on November 5. The low-resolution COA for that day was 11.2 µg m$^{-3}$. On the other hand, the low-resolution analysis tends to underestimate the COA during

most of the days with COA levels below 2 µg m$^{-3}$ resulting in a relatively small overprediction (25% versus 20%) of its average contribution to OA during the full period.

The BBOA concentration was overestimated by the low-resolution analysis by 20-30% at days with high concentrations (BBOA above 5 µg m$^{-3}$) (Fig. 10). The highest discrepancy was observed on January 2, which was the day with the highest BBOA concentration in the five-month period (13 µg m$^{-3}$). The low resolution BBOA was 1.8 times higher than the high resolution at that day. At the same day the low-resolution LO-OOA was underestimated (9.7 µg m$^{-3}$ for the high-resolution results and almost zero for 24 h). These discrepancies were quite systematic ($R^2$=0.95) and resulted in an overestimation of the BBOA for the full period by the low-resolution approach (24% of the OA versus 16% for the 30 min analysis). So at least for this dataset the use of the daily resolution leads to a systematic overestimation of the BBOA during most days.

Despite the discrepancies, the results of the two approaches for the primary factors were relatively well correlated with the $R^2$ varying from 0.82 for the COA to 0.96 for the HOA. This was not the case with the secondary factors where the $R^2$ between the results of the 30 min and 24 h analysis for the 127 daily data points was 0.24 for the LO-OOA and 0.32 for the MO-OOA. This underlines the sensitivity of the LO-OOA/MO-OOA split to the temporal resolution used for the analysis. Hildebrandt et al. (2009) argued that the two OOA factors often appearing in such analyses represent roughly the upper and lower limits of OA aging encountered during the study period. Averaging of the measurements while moving from 30 min to 24 h samples is therefore expected to limit the range of OA states encountered and therefore to bring closer to each other the LO-OOA and MO-OOA factors resulting from the PMF analysis. Use of data from more seasons in the analysis will introduce significant uncertainty because of the different dominant chemical processes leading to SOA production and also the chemical aging mechanisms of primary OA (Canonaco et al., 2013; Kaltsonoudis et al., 2017). Despite these discrepancies for the daily results, the averages for the study period were relatively consistent (less than 5% of the OA) for the two analyses. The 24 h MO-OOA contribution was 15%, while the 30 min was 18%, and the 24 h LO-OOA contribution to the total OA was 13% while in the 30 min analysis it was 17%.

In order to examine further these discrepancies, we have analyzed separately the low and high OA concentration days. We sorted the dataset and spit it in two halves, one with the low and the other with the high concentration days. We then compared the results of the low and high temporal resolution PMF for these two subsets. During the low concentration days, as expected, there were higher fractional discrepancies than during the high concentration days (Figs. S22-S23). The 24 h COA, HOA and MO-OOA mass concentrations were in general lower when compared with the 30 min results during low concentration days, while BBOA and LO-OOA were higher. There were also a few days in which the 24 h results indicated zero COA, while the 30 min COA mass concentration was around 1 µg m$^{-3}$. The COA signal in these days was allocated by PMF to all other four factors, including primary (HOA and BBOA), but also the secondary factors (MO-OOA and LO-OOA). The primary to secondary split changed relatively little as there were changes also in the secondary factor contributions. Days with zero HOA mass concentration were present in the low temporal resolution results. The 30 min results for these days showed HOA mass concentrations below 1 µg m$^{3}$. These results suggest strongly that the low temporal resolution PMF results provide estimates of the average contribution of the various source to the total OA for longer periods (a few months) that are consistent with the high temporal resolution PMF results. However, for specific days and especially for low concentration periods the discrepancies of the low and high temporal resolution results can be quite high.

A bootstrap analysis has also been performed in order to characterize the uncertainty of the PMF results. The average estimated concentration of each factor to the total observed OA varied by less than 30% of its mean value (Fig. S19). The lowest difference was observed for MO-OOA (12%) and the highest for BBOA (30%).

The factor profiles for the low and high temporal analyses are compared in Fig. 9. The spectra for HOA (7°), MO-OOA (11°) and COA (13°) are quite consistent with each other and appear to be less sensitive to the temporal resolution of the analysis. On the other hand, there are significant differences in the spectra for BBOA (22°) and LO-OOA (30°) that are clearly a lot more sensitive.

## 3.5 Analysis of month-long datasets

355 Our analysis so far has focused on the effects of the temporal resolution for the full data set, that is all five months of a period with similar characteristics and sources together. For the 24 h analysis there were 127 samples using for the PMF. Certain field campaigns last a lot less than five months, so in this section we examine the analysis of the data of each month separately from the rest. For the 24 h analysis, this involves 28-31 samples for November to February and only 17 samples for March (due 360 to missing data for technical reasons). March is an interesting test case because the set of data is probably too small for such a PMF analysis. A five-factor solution was obtained for each month in the 24 h resolution analysis.

The primary/secondary split calculated for the four months with complete data sets for the 24 h resolution was quite consistent (differences less than 10% of the OA) (Fig. 11). For March, the 365 existence of only 17 data points resulted in significant error, as expected, with the 24 h resolution analysis estimating that 34% was primary, while the 30 min analysis estimate was 58%. So the 30 data points appear to be sufficient in this case, but the 17 produced erroneous results at least for the March conditions.

The results for the monthly average contribution (using only the 28-31 data points) of the 370 various factors are quite encouraging (Fig. 12). The estimated BBOA contribution differed by 5% or less. The differences in HOA were a little higher, but still less than 10% of the total OA. Little higher discrepancies were found for COA especially in November and December. For example, during November the COA was 23% of the OA according to the 30 min analysis and 34% based on the 24 h results.

375 The estimates of the contributions of the secondary factors had the highest discrepancies differing by as much as 13%. In general, there was better agreement for the sum of the OOA factors than for their individual values. For the MO-OOA the highest discrepancy was observed in December, in which the two resolution results differed by 10% (20% for the 30 min results and 10% in the daily resolution). The LO-OOA highest difference was 13% and was found for January during which the 380 30 min LO-OOA was 17%, while the 24 h was 30% of the OA.

During March the use of the small dataset with 24 h resolution, resulted in significant underprediction of the BBOA, underprediction of the COA and significant overprediction of the MO-OOA (Fig. 12). This indicates that a 24 h sample size of 17 days will result in significant errors in five factor solutions in periods like March which is also at the end of the heating period.

## 4 Conclusions

In this study the impact of the time resolution in the PMF results of an ACSM was examined using data from an urban site in Athens during a relatively cold period. During this period the OA had both primary sources (transportation, biomass burning and cooking), but also a significant secondary component. Analysing the full data set (127 days) together, the same number of factors were found for each data averaging interval (30 min, and 1 , 2 , 4 , 6 ,8, 10, 12, 24 h): three primary and two secondary.

The average contribution to the total OA of each factor varied within 8% between the lowest and the highest temporal resolution results. This suggests that even the lowest resolution of 24 h samples often used for offline AMS analysis can provide valuable insights about the secondary OA and the major sources of the primary OA for a multi-month period. The improvement of the results going from 24 h to 12 h was marginal, suggesting that for at least this five-month period little would be gained by doubling the number of samples from 127 to 254.

The highest discrepancy between the 30 min and 24 h analysis results was found for BBOA. The low temporal analysis overestimated BBOA by 8% of the OA (24% for the 24 h versus 16% for the 30 min). One should note that the same difference can be viewed as a 50% overestimation of the BBOA contribution. However, the accuracy of even the 30 min estimates is expected to be low, so this 50% overestimation may be misleading. The tendency of the 24 h analysis to overestimate BBOA in this dataset is noteworthy and should be compared with the results of similar analysis in other locations. The discrepancies for the other OA components were an underprediction of the HOA by 6% of the OA by the 24 h analysis (23% versus 29% for the 30 min), an overprediction of the COA by 5% (25% versus 20% for the 30 min), an underprediction of the LO-OOA by 4% (13% versus

17%) and an underprediction of the MO-OOA by 3% (15% versus 18%). The tendency towards a small underprediction of the secondary OA should also be examined in future studies.

The discrepancies between the results of the 24 h and the 30 min analysis increased when the PMF analysis was performed for just one month (28-31 days) assuming that only one month of daily filter samples was available for off-line AMS analysis. However, the differences of the estimated contributions of the various factors remained below 13% of the total OA. This suggests that even one month of daily samples can provide valuable insights about the OA components and sources on average. Of course, the uncertainty is higher compared to multiple-month data sets.

The uncertainty of the off-line AMS analysis will be a lot higher if one focuses on individual days, even if there are months of available data. Discrepancies of a factor of two were observed for several factors when the 30 min and 24 analyses were compared. These high discrepancies were observed not only for relatively clean days, but some of the days with the highest concentrations of the BBOA and COA factors. This suggests that the off-line AMS results are quite uncertain for specific days. One should note here that unit mass resolution ACSM data were used in our study. The uncertainty of the off-line analysis for individual days may be lower if high mass resolution AMS measurements are used for the off-line analysis.

Finally, the factor profiles determined by the 30 min and 24 h resolution analysis were relatively similar for HOA (theta angle 7°), but there were some differences for MO-OOA (11°) and COA (13°). There were significant differences in the spectra for BBOA (22°) and LO-OOA (30°).

*Data availability.* Measurement data are available by request by nmihalo@noa.gr.

*Supplement.* The supplement related to this article is available at:

*Author contributions.* CV and SNP designed the study and wrote the paper. CV did the corresponding PMF analysis presented here. IS and NM obtained and provided the ACSM data. All authors contributed to the interpretation of the results and edited the manuscript.

*Competing interests*. The contact author has declared that neither they nor their co-authors have any competing interests.

*Financial support.* This work was supported by the PANhellenic infrastructure for Atmospheric Composition and climatE chAnge (PANACEA) project (grant no. MIS 5021516) and the IMSAP
project (grant no. TIEΔK 03437) of the Greek General Secretariat for Research and Innovation.

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

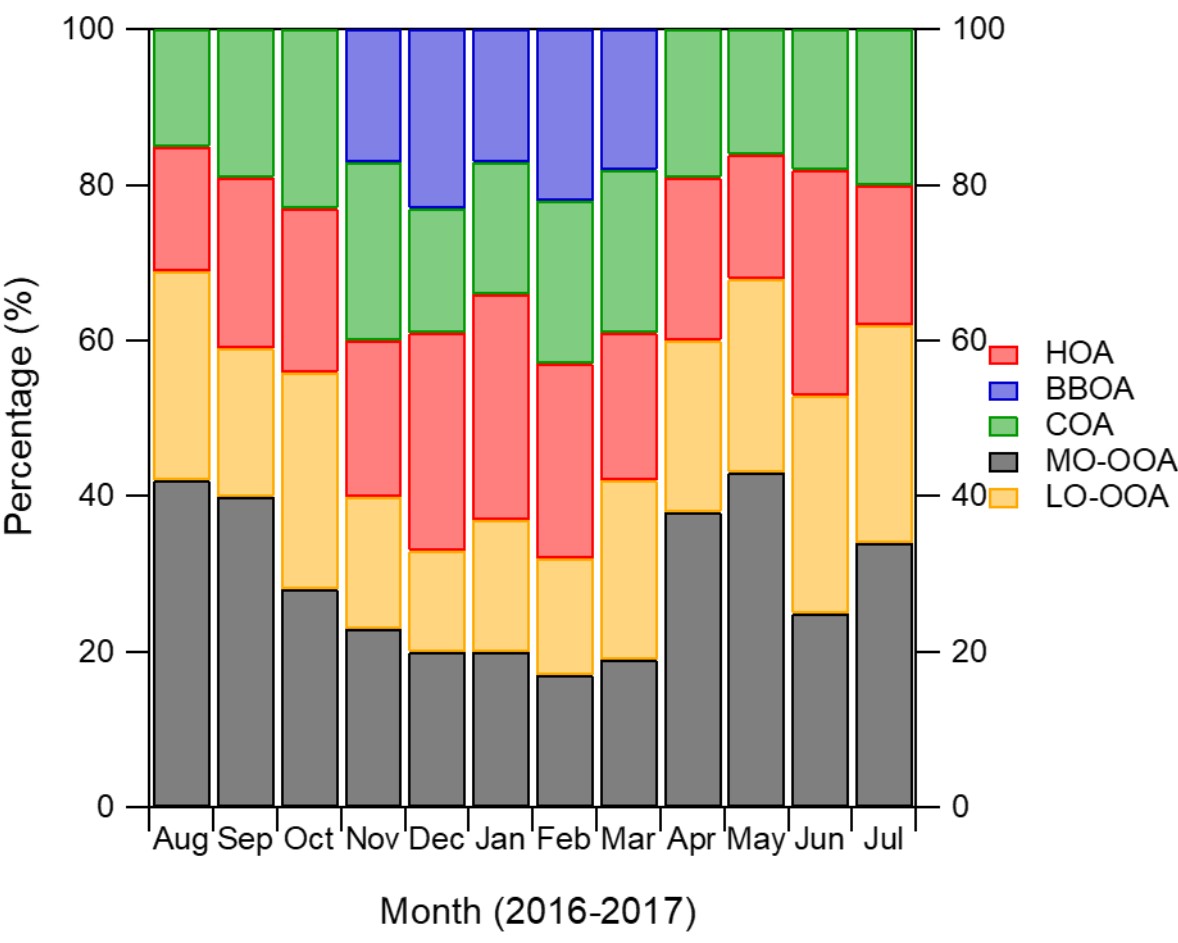

630

**Figure 1:** Fractional contribution of each factor to the total OA for the PMF analysis of each month separately. The 30 min data set was used for this analysis.

635

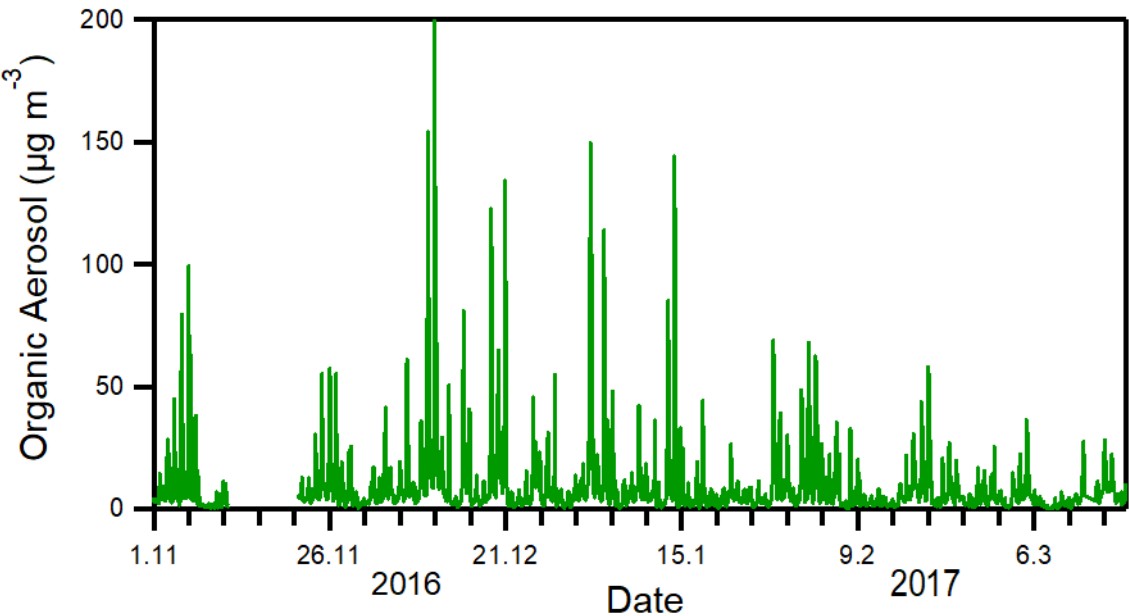

**Figure 2:** Organic aerosol concentrations measured by the ACSM in the center of Athens for the
November-March cold period analyzed in this work. The time resolution is 30 min.

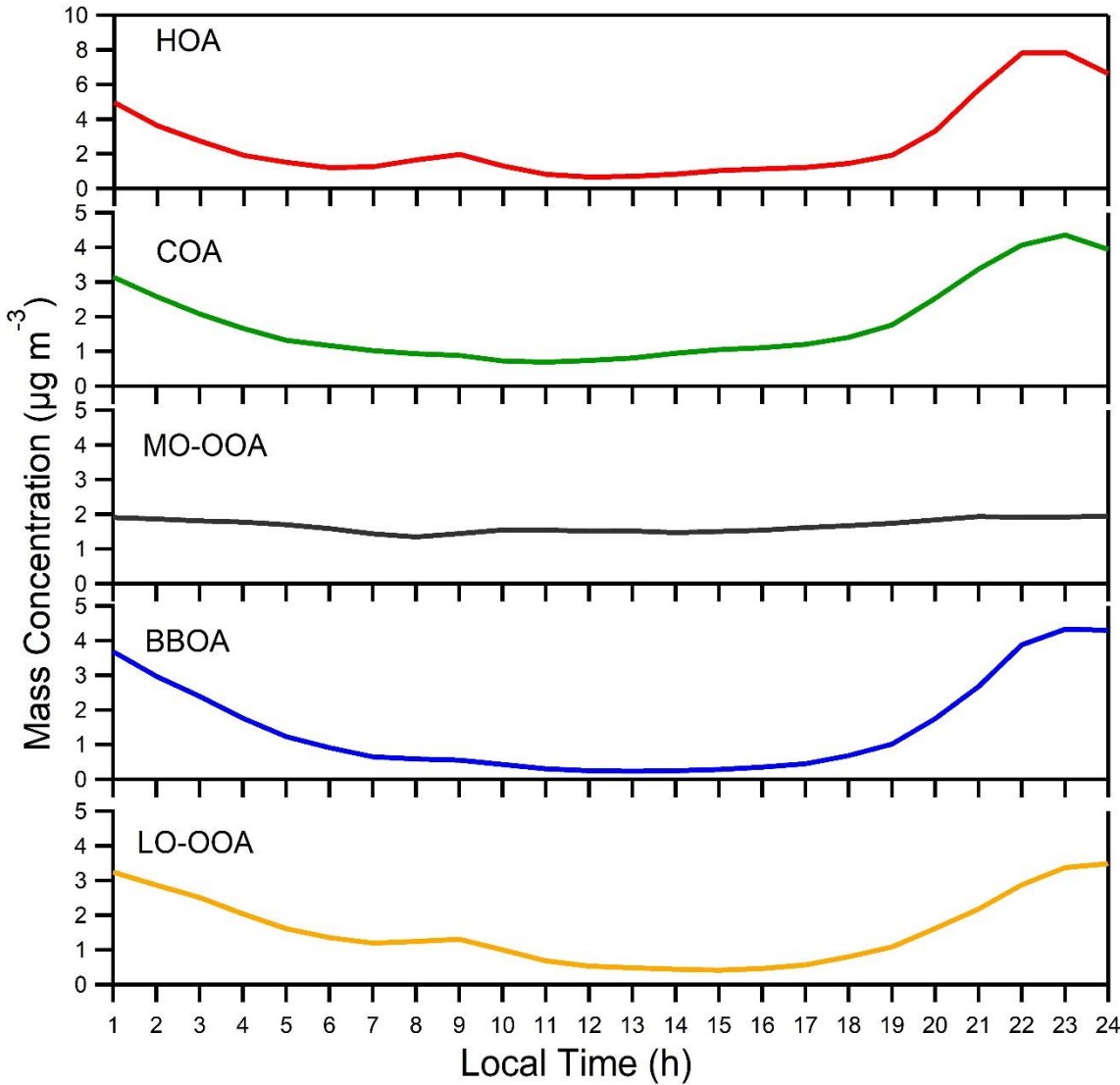

660

**Figure 3:** Average diurnal profiles for the five factors derived from the 30 min time resolution PMF results during the cold period (November 2016- March 2017). Different scales are used for the HOA and the rest of the OA components.

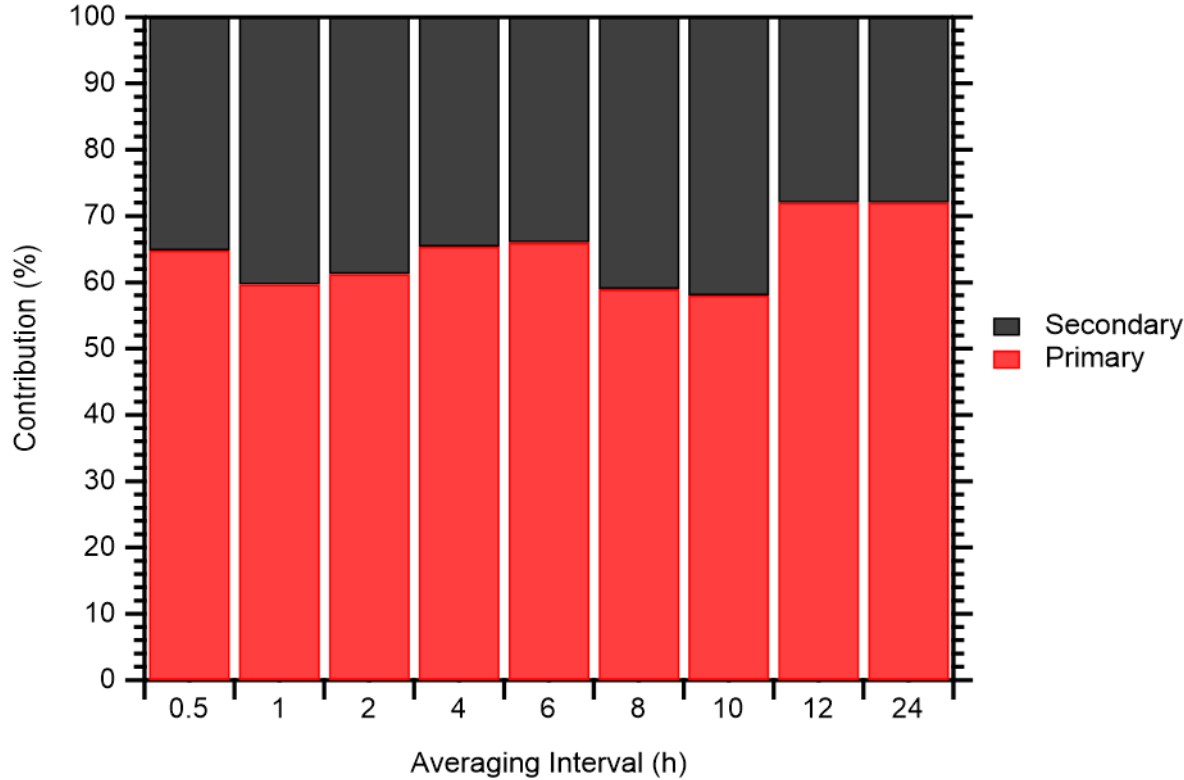

**Figure 4:** Contribution of the sum of primary factors (HOA, COA and BBOA) and the sum of secondary factors (MO-OOA, LO-OOA) to the total OA for the various averaging intervals for the cold period.

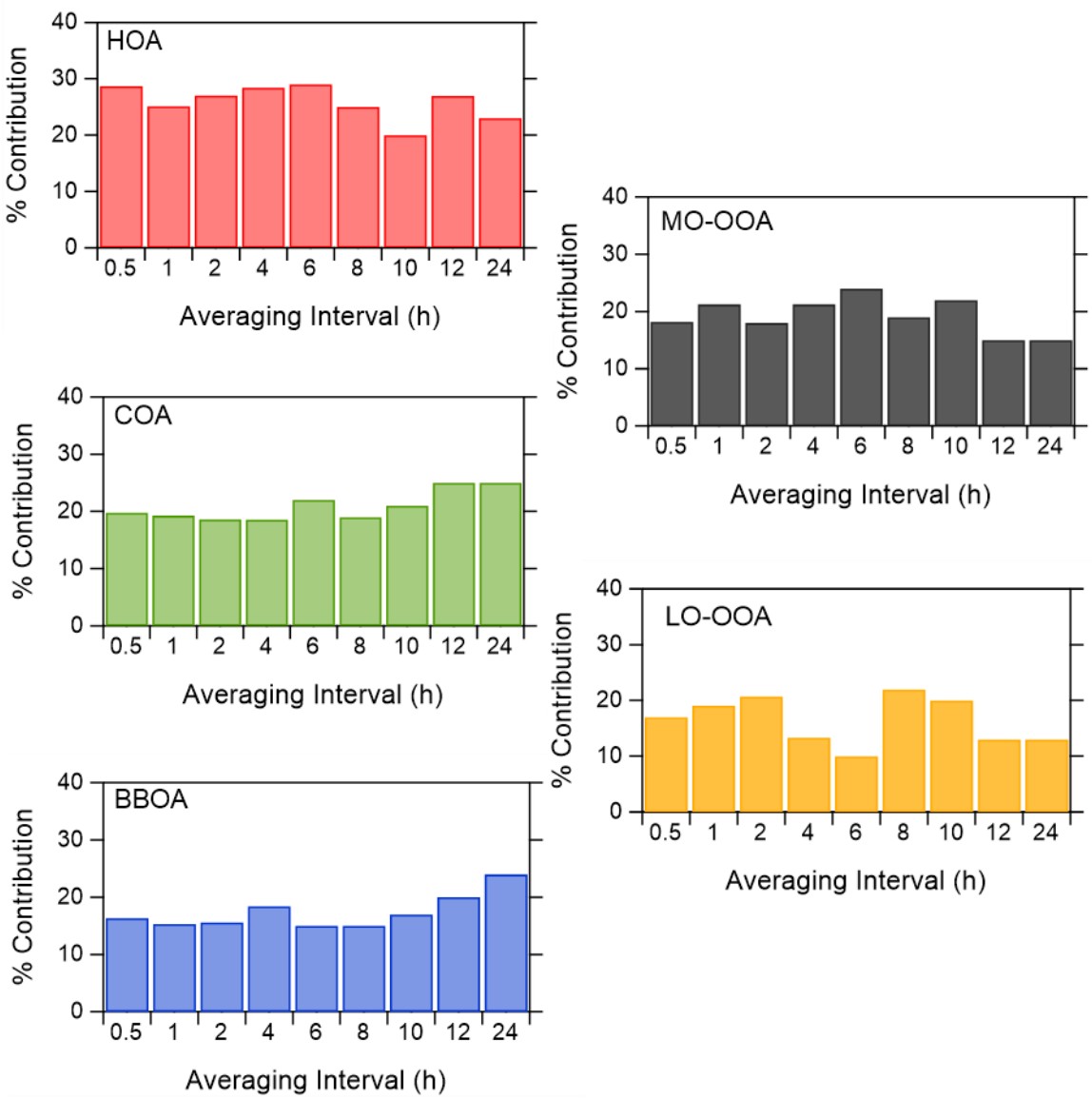

680

**Figure 5:** Contribution of each factor to the total OA for the different time resolutions.

685

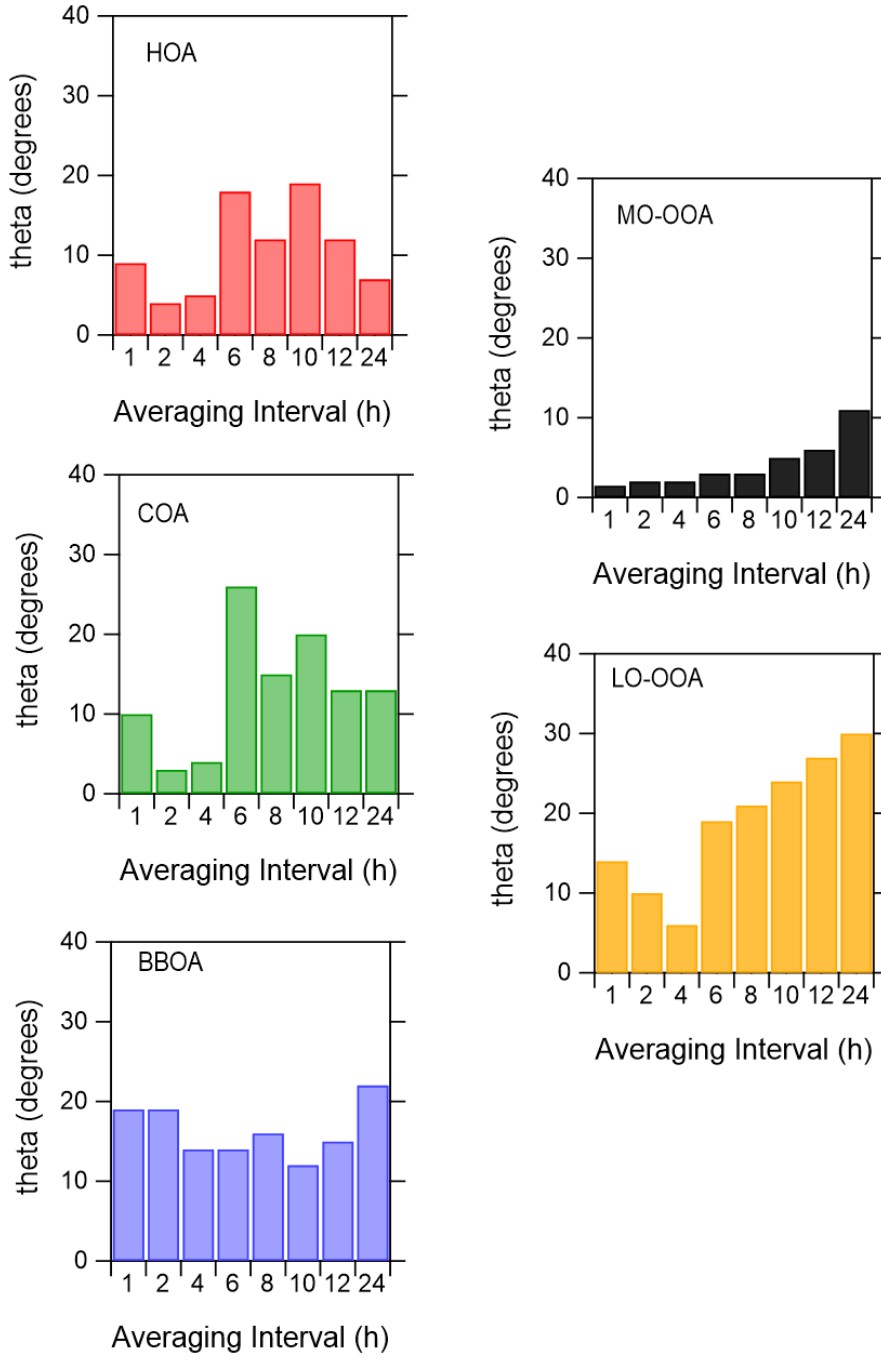

**Figure 6:** Theta angle between the spectra derived from the 30 minutes PMF analysis and the spectra derived from the analysis at different time resolution.

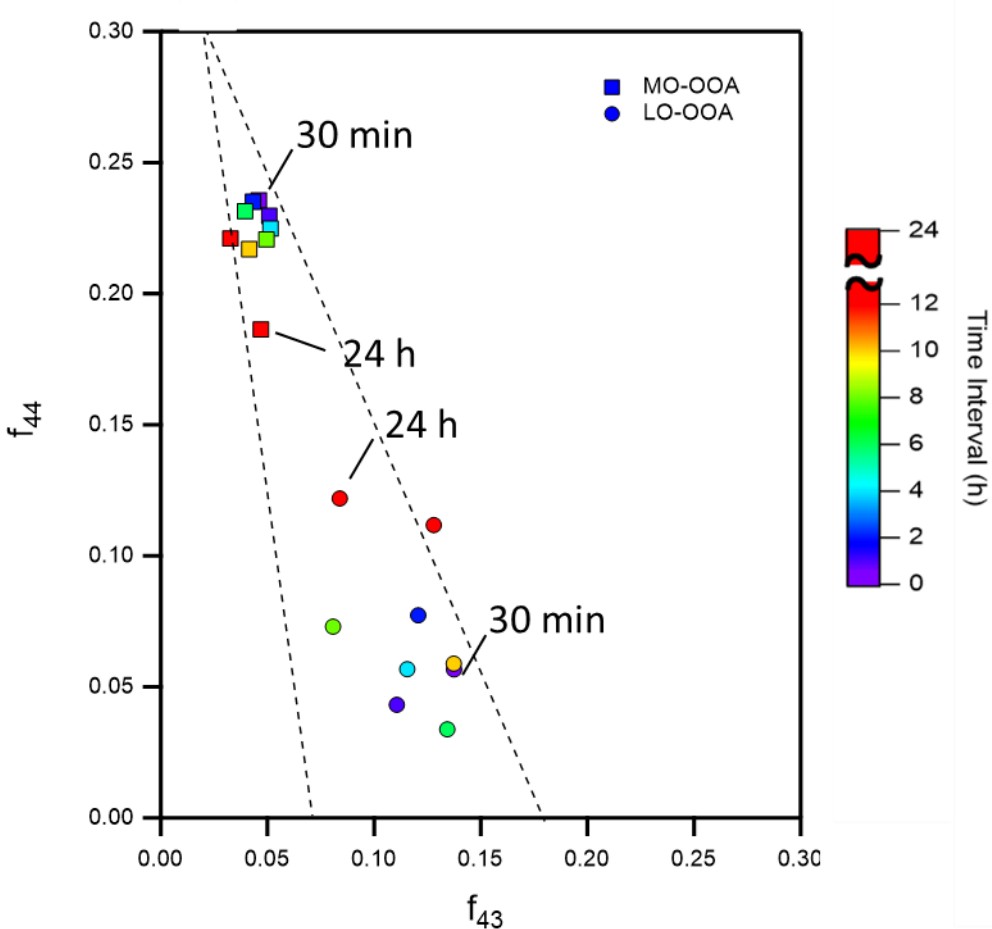

**Figure 7:** *f44* vs *f43* triangle plot of the two secondary factors for the different time intervals of the PMF analysis. The LO-OOA is shown with circles and the MO-OOA with squares. The results of the 30 min and the 24 h analysis are highlighted.

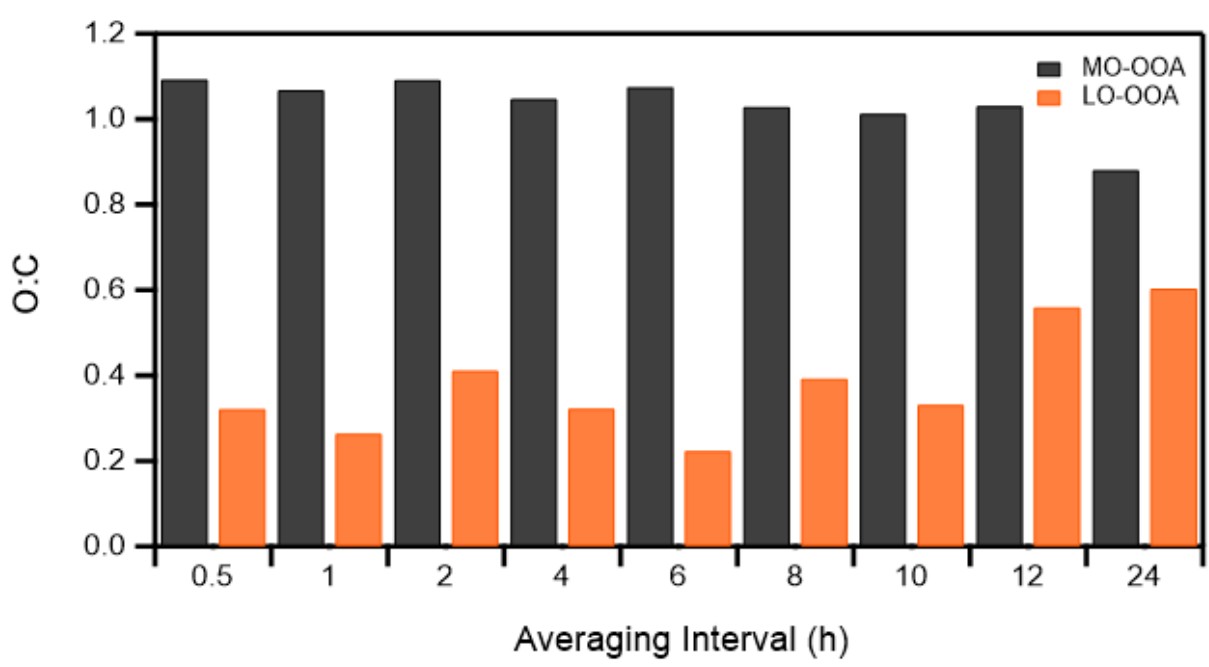

**Figure 8:** Atomic oxygen to carbon ratio (O:C) of the two secondary factors (LO-OOA and MO-OOA) for the different temporal resolution tested in this study.

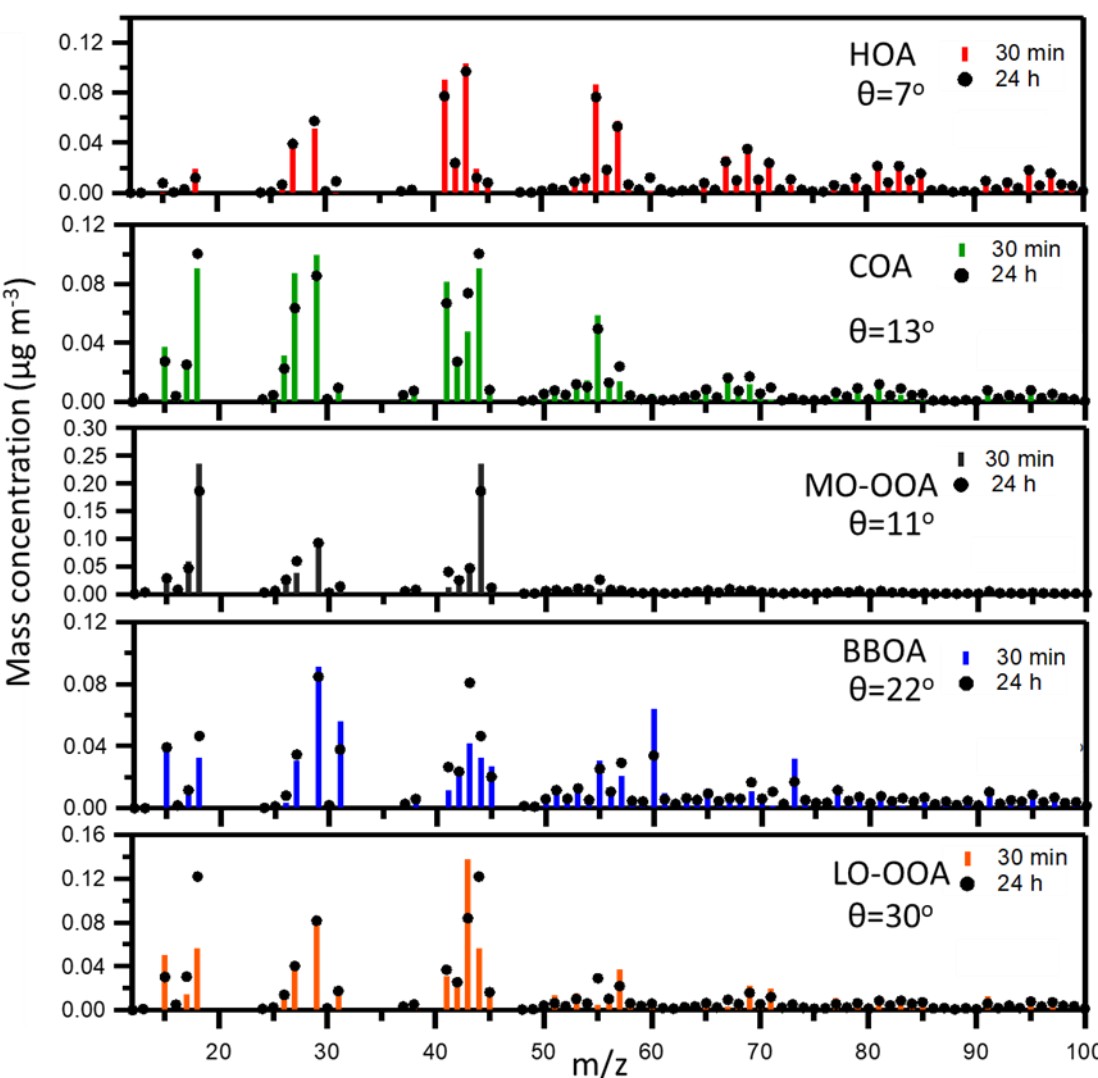

**Figure 9:** Comparison of the spectra of the factors in the PMF analysis of the 30 min and the daily temporal resolution data. The theta angle of the corresponding vectors/spectra is also shown. Different y axes are used.

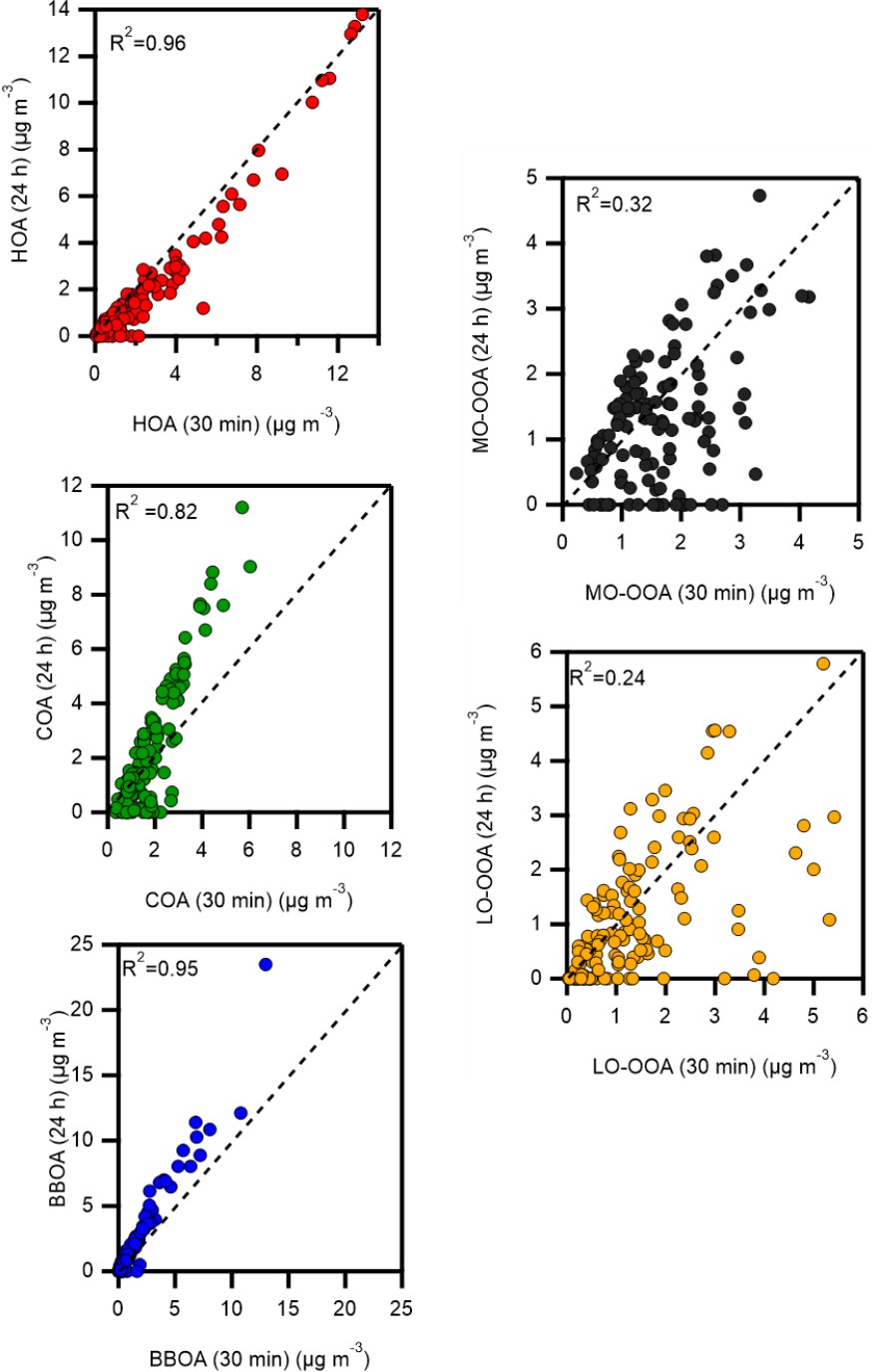

**Figure 10:** Comparison between the results of the 24 h analysis and the daily averages of the 30 min analysis for each factor for the cold period. The 1:1 lines are shown. Different axes are used.

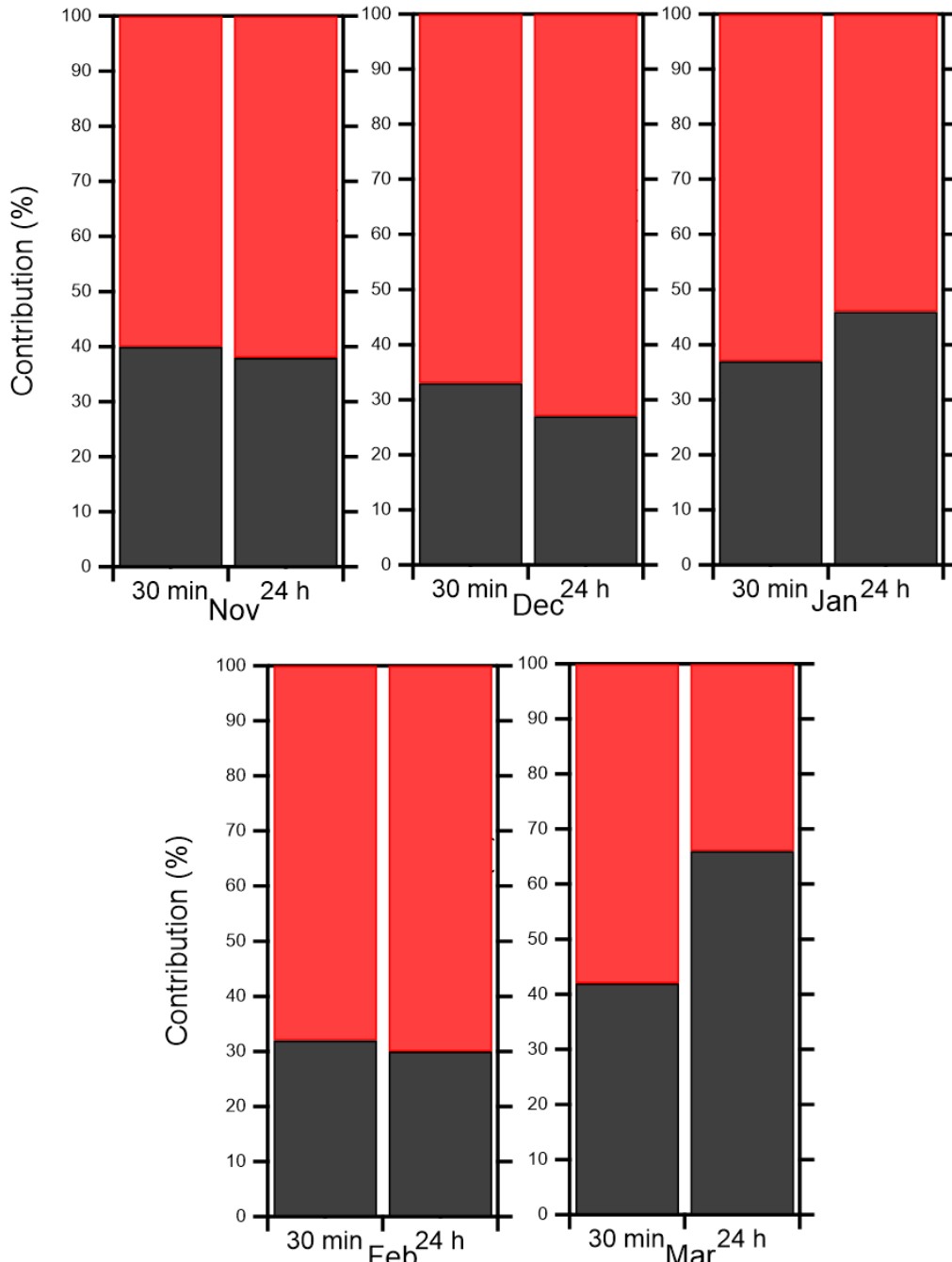

**Figure 11:** Primary (red) and secondary (black) contribution to the total OA for the high and the low time resolution results for each month.

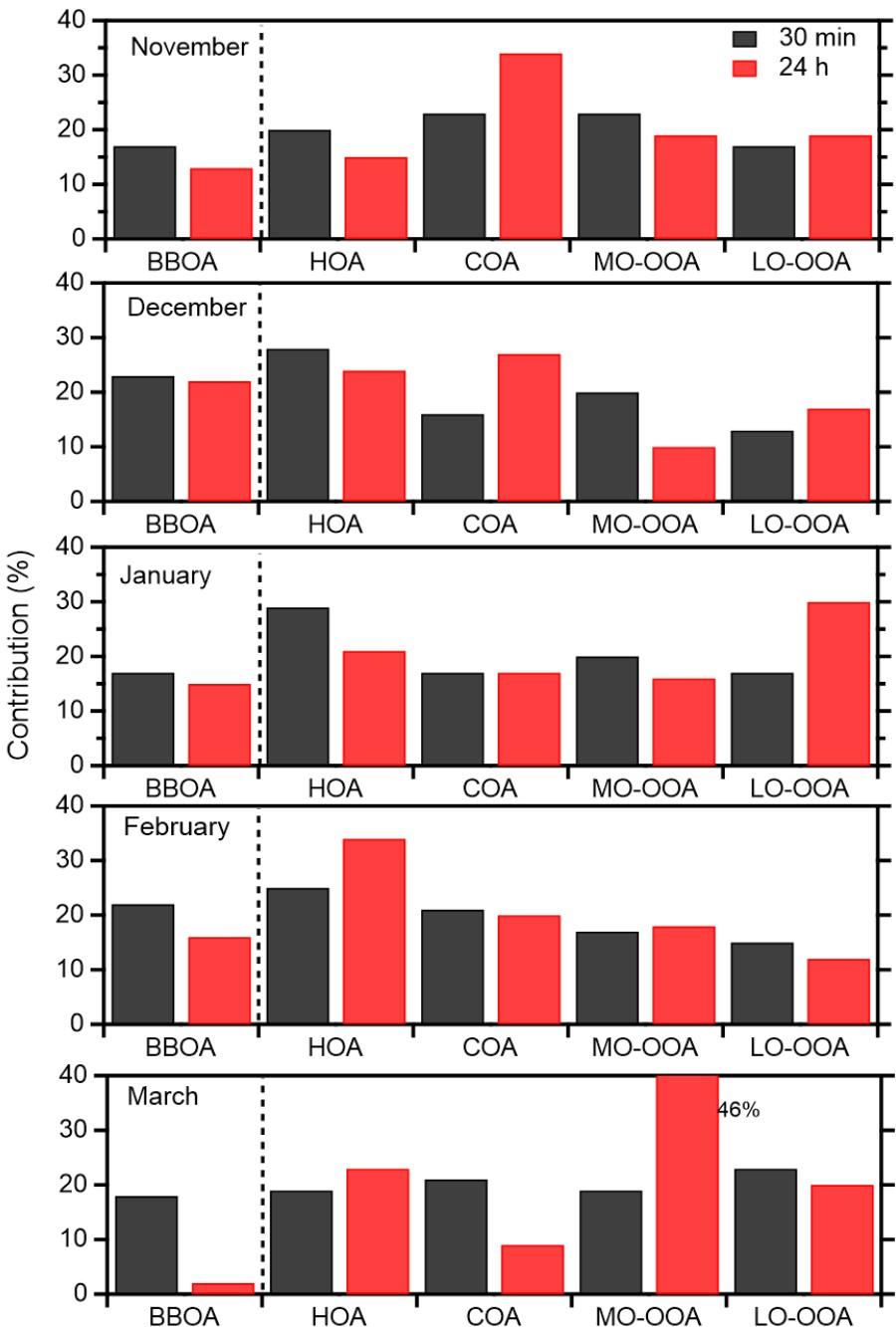

**Figure 12:** High and low time resolution contribution of each factor to the total OA for each month individually.