# Peer review of "The Effect of the Averaging Period for PMF Analysis of Aerosol Mass Spectrometer Measurements during Off-Line Applications"

_Atmospheric Measurement Techniques, 2022_

## Author Response (AR1)

**Responses to the comments of the Reviewers**

**Reviewer 1**

(**1**) Thanks to authors for raising an important issue via this article. This is completely true that Off-line AMS measurements have added valuable information to OA chemical speciation and sources when the use of online AMS is not possible especially for the remote sites. However, these offline measurements have low temporal resolution and how this affects the performed source apportionment, it has not been discussed thoroughly in the research community. Results showed here the sources apportionment performed on low- and high-resolution data retrieves the same number of factors, however their contribution to the total mass is relatively different. Even the resolved factors from the low-resolution data introduces more error in the factor profiles. Results are interesting, however I found more discussion is needed for some parts. I have listed the major concern below:
We appreciate the positive assessment of our work by the referee. Our responses (in black) and the corresponding changes in the manuscript follow each comment of the reviewer (in blue).

(**2**) Method section, as I understood PMF was performed on the different resolution dataset. So basically, origin high resolution data has been averaged to do this exercise. However, I am wondering, what about uncertainty. The uncertainty matrix is automatically generated for the original data, but then how it was processed for the different time resolution? Did author average the uncertainty for different resolution? Also, is the simple average has been used or weighted average? While averaging, how any spikes or specific events are treated because they all would affect the source apportionment. I would recommend to elaborate this in the manuscript.
To avoid unnecessary complications, the uncertainty was simply averaged for the different temporal resolution datasets in the main analysis. Following the advice of the reviewer, we have performed an additional sensitivity test using a geometric average for the calculation of the error matrix of the 24 h results. The arithmetic average was used for the AMS measurements themselves in this test. In this rather extreme test, the predicted contribution of each primary factor to the total OA changed by less than 10% compared with the results using the arithmetic average error. The highest discrepancy was observed for MO-OOA and was 16%. Once more the resulting AMS spectra from the PMF showed higher discrepancies. The results of this sensitivity analysis are now discussed in the revised paper.

Given that the high temporal resolution dataset has a 30 min temporal resolution the relatively high concentration data points ("spikes") were kept in the dataset to avoid losing information about the sources of primary organic aerosol. To explore the effect of these periods on our results, we have added to the paper a more detailed analysis of seven days with the highest observed 30 min OA concentrations (above 100 μg m$^{-3}$) during the five-month period examined in this work. The high OA concentrations were observed at nights, between 21:00 and 3:00 and remained high for several hours. The

results of the comparison of the 24 h and 30 min results during these days with high concentration periods were quite consistent with the rest of the days for the primary OA components. So, we did not observe a notable change in behavior of the PMF using low temporal resolution data during these interesting high concentration events. This analysis and discussion have been added to the revised paper (please see also our response to Comment 6 of Reviewer 2).

**(3)** The variation in the spectra of various factors has been evaluated using theta angle. Author have referred a paper for this, but it would be nice to provide some details in the given paper. Also, the discussion based on the theta angle is minimal, what about comparing with other approaches to check the variation?

We have followed the suggestion of the reviewer and added the requested details about the theta angle. We have also added the $R^2$ values as an alternative comparison metric of the mass spectra in the revised paper.

**(4)** Another concern is the discussion on the percentage change in the contribution of different factors. Yes, I can see a change in the contribution of primary factors to the total OA from 65% to 72% when the resolution moved from 30 min to 24 hours. Is that change in the contribution significant? What about the averaging error? Or any other errors i.e. PMF error? I can't find any detailed discussion on this in the manuscript.

Considering the uncertainty of the source apportionment approaches like PMF, a change of a source contribution from 65% to 72% is of secondary importance. In order to investigate more the uncertainty of our PMF results, a bootstrap analysis of the 24 h results has also been performed. The results showed that the mass concentration of each factor could differ by as much as 30% of the factor concentration from the mean value. Following the suggestion of the reviewer we have added more information about the errors and their significance including the results of the bootstrap analysis to the manuscript.

**(5)** Authors have provided the comparison between 30 min and 24-hour factor profiles, and differences between them. But why these differences have been found, what could be the reason for such discrepancies, no discussion on that. Like why COA tends to over predict when it is below 2 $\mu g/m^3$?

We have added the recommended discussion of the potential reasons for these differences. The most important one is probably the reduction of information provided to PMF when one moves from thousands of measurements (for the 30 min dataset) to a little more than one hundred (for the 24-hour one). Given that the diurnal variation of the source contributions is lost during this averaging it is quite surprising that the differences that we found are that low. One of our conclusions though is that the change in factor profiles is higher than that of the source contribution. This suggests that some of the changes in the profiles are at m/z values which are less important for the quantification of the contribution of each factor. At low factor concentration levels, the errors are expected to be much higher. For example, there are several days (15 out of 127) during which a zero COA concentration is estimated in the 24-hour analysis. The

COA during these days is assigned to a subgroup of the other four factors. Different factors were overestimated in each one of the 15 days in which the COA mass concentration was zero. Similar errors were found for the other primary factors in specific days. These issues are now discussed in the revised paper (please see also response in Comment 2 of Reviewer 2).

Minor Comments

**(6)** Please add the reference for HOA mass spectrum.
A brief discussion of the assignment of the factor profiles to specific sources for the high temporal resolution data set and the corresponding references have been added to the revised paper.

**(7)** What about COA diurnal? I would suggest to add that information.
The average diurnal COA profile is shown in Figure 3. A brief discussion about its behavior has been added.

**(8)** Earlier in the manuscript, degree symbol has been used but later it is written degrees, please be consistent.
The word "degrees" has been replaced by the corresponding symbol in all the paper.

**Reviewer 2**

**(1)** The manuscript titled "The Effect of the Averaging Period for PMF Analysis of Aerosol Mass Spectrometer Measurements during Off-Line Applications" aims to understand how differently time resolved measurements affect the source apportionment results. This is an important problem to tackle as it will allow for systematic comparisons between online and offline source apportionment results and their effectiveness in interpreting real-world source contributions. As the authors pointed out, developing this understanding is especially important since it is not always possible to make online measurements. The paper is well written in terms of language and the analysis is nicely structured. I am also happy with their input protocols for SoFi (lines 86-95). I do have some major concerns however, which are primarily rooted in data treatment/processing and explanation of causatives for observations. I think the paper needs to undergo major revisions before it could be considered suitable for publication. My comments are listed below:

We appreciate the helpful suggestions and comments. Our responses (in black) and the corresponding changes in the manuscript follow each comment of the reviewer (in blue).

**(2)** One of my major concerns is that the paper in its current form reads like a commentary on what percentage variations where observed while running SoFi on differently averaged datasets. However, no insights are provided on likely factors controlling the variations. I see this as a little bit of a drawback since it is not helping the reader in using this paper as a reference in interpreting their own results. For example, in lines 174-177, COA is said to have an opposite behavior to HOA in low temporal resolution analysis. COA is overestimated during high COA concentration periods and underestimated during low concentration periods. However, no informed speculation is made for why this should be happening, which can help the reader. I am guessing that this may be due to SoFi allocating more fraction of the total signal to other factors when data-to-error ratio is lower during low concentration periods.

We have followed the suggestion of the reviewer and we have analyzed separately low and high concentration periods. The low concentration periods showed, as expected, higher fractional errors than the high concentration ones. The 24-h COA, HOA and MO-OOA mass concentrations were underestimated when compared with the 30 min results at low concentration periods, while BBOA and LO-OOA were overestimated. There were even days in which the 24-h results indicated zero COA, while the 30 min COA mass concentration was around 1 μg m$^{-3}$. Also, days with zero HOA mass concentration were observed for the low temporal resolution results. The 30 min results for these days showed HOA mass concentrations below 1 μg m$^{-3}$. An analysis of the low and the high concentration periods separately has been added to the paper with the corresponding figures to provide insights about the reasons of the discrepancies of the high- and low-resolution analyses. The low temporal resolution results can give a good estimate of the sources contribution to the total OA for longer periods (a few months) but for specific days and especially for low concentration periods the low temporal resolution error can become significant.

**(3)** Another concern is regarding the data averaging methods employed in this study. The way PMF results would pan out depends on how the data was averaged across different time periods. The authors should discuss whether it was arithmetic or geometric average or weighted average etc. and how it influenced SoFi inputs. SoFi would produce robust results for high data-to-error ratios (e.g. high concentration periods) separating them more clearly in bootstrap tests. However, depending on how the peaks were handled during averaging, the data-to-error ratios could decrease influencing SoFi outputs and making them more uncertain. Given Figure 2, which shows many high concentration events for organic aerosol, a discussion on the averaging technique is very important.

The arithmetic average was calculated and used in our analysis both for the data matrix and the error matrix for the different temporal resolutions. Given the importance of this averaging especially for the error, a sensitivity test using a geometric average for the 24 h error matrix has been performed. The results of the test indicated that the change in the estimated contribution of each primary factor to the total OA was less than 10% compared with the PMF results using the arithmetic average of the errors. The highest discrepancy was found for the secondary factor MO-OOA and was 16%. Once more the resulting AMS spectra from the PMF for the various factors showed higher discrepancies. The results of this sensitivity analysis are now discussed in the revised paper. High concentrations were not handled differently from the rest of the data. This is now made clear in the manuscript.

Following the suggestion of the reviewer a bootstrap test has been made to quantify the uncertainty of the PMF analysis. The results indicate that the average estimated concentration of each factor to the total observed OA varies by less than 30% of its mean value. A discussion of the above issues has been added to the revised paper.

**(4)** Authors show variations in factor contributions broadly in the range of 6-15% across their analysis. Following from (2), I am curious about how significant are these numbers because they do not seem too large. This is encouraging for offline analysis. Could the differences be explained by error propagation in the averaging technique employed? The paper would benefit from a more thorough discussion on this. I recommend running a sensitivity analysis on this matter comparing PMF results from 30-min original data with results obtained for just the 24-h averaged using different averaging techniques, or at least discussing potential effects in the text.

Following the reviewer's comment, a sensitivity test using a geometric average for calculating the 24 h error matrix has been performed. The results of the two PMF tests differed by less than 16% for the average contribution of all factors. The estimated AMS spectra of the various factors showed higher discrepancies. For example, the COA theta angle between the geometric and the arithmetic average error results was 30 degrees. The best agreement was observed between the two MO-OOA spectra (8 degrees). The results are now discussed in the revised paper (please see also our responses to Comments 2 and 3 of Reviewer 1).

**(5)** In lines 154-159, the authors mention O:C ratios of MO-OOA and LO-OOA to be moving toward each other as the time resolution decreased. This seems to me an

evidence of the averaging effect on the data. SoFi could separate the two better for higher time resolution measurements since the data captures temporal dynamics in concentrations resulting from ambient oxidation chemistry more clearly telling apart more from less oxidized. Consequently, SoFi can comfortably separate such contributions into their own factors. Is this reasonable thinking? It would be good if the authors shed more light on the O:C trends they noticed for the secondary components of their measurements.

The two secondary factors were separated due to their differences in specific *m/z's*, like 43 and 44, but also due to their different atomic oxygen to carbon (O:C) ratios. At high temporal resolution, the two factors can be better separated from each other by PMF. On the contrary, in low temporal resolution measurements, a mixing of the two secondary factors is noticed. In this dataset the difference between the O:C of the two oxidized factors (MO-OOA and LO-OOA) decreased as the temporal resolution was getting lower. This was also evident in the $f_{44}$ vs $f_{43}$ triangle plot, in which the two factors were approaching one another, as the temporal resolution was decreased. A brief discussion and a new figure with the O:C of the MO-OOA and LO-OOA factors for the different time resolutions has been added to the revised paper.

**(6)** How different would be a 24-hr filter sample data from a 24-hr averaged high time resolution online measurement? The authors should at least discuss the challenges involved with replicating a 24-hr offline measurement this way. For example, in mathematical treatment of high resolution data, very short term concentration outliers (peaks or drops over few hours) can drag averages up or down to some extent in replicating lower resolution. However, such real-time, short-term concentration peaks or drops may not have the exact same impact on a filter sample being collected over a 24-hr period. Some insights on this issue would be helpful for a reader.

We agree with the reviewer that there are some experimental issues which can lead to differences between actual 24-hour filter samples with the 24-hour averaged online data discussed in this work. These issues include the blank uncertainty, the sample extraction efficiency, potential filter sampling artifacts, etc. Our objective in this work was to quantify the effect of the use of low-temporal resolution data. We have added a brief discussion of these experimental issues in the revised paper, and we plan to address them in follow-up experimental work.

In the revised paper, we analyze in more detail the days with the highest observed OA concentrations, which could behave as outliers in our work. We focused on the days with the seven highest 30 min total OA periods ("spikes") during the five-month analysis period. The high OA concentrations (above 100 μg m$^{-3}$) were observed at nights, between 21:00 and 3:00 and remained high for several hours. The comparison of the 24 h and 30 min PMF results showed that the PMF analysis had similar behavior in these high concentration days as on average and this is rather encouraging for the off-line AMS analysis. A discussion of these additional results and their implications has been added to the paper.

**(7)** I like figures 5 and 6 a lot because they give a very nice overview of the results of this paper. It is very interesting that while % contributions for most individual factors appear to be within 2 standard deviations from average across the different time

intervals, the theta angles are considerably different in some cases (e.g. HOA, COA and LO-OOA). Now, the primary to secondary component split is pretty similar across the different averaging intervals. So, looking at HOA, COA and BBOA in Figure 6, does this mean that the aerosol signal is being differently allocated by SoFi between the primary components for different averaging intervals?

The situation appears to be more complex, and the differences change depending on the specific day examined. For example, for a few days the low temporal resolution (24 h) analysis estimated zero contributions of COA, while high resolution (30 min) estimate of COA was around 1 μg m$^{-3}$. The COA signal in these days was allocated by PMF to all other four factors, including the primary (HOA and BBOA), but also the secondary factors (MO-OOA and LO-OOA). The primary to secondary split changed relatively little, but there were changes also in the secondary factor contributions. Additional analysis and discussion of these results has been added to the manuscript (please see also our response to Comment 5 of Reviewer 1).

We agree with the reviewer that average differences of the estimated contribution of each factor to the total OA for the different time resolution results are small. However, the corresponding changes of the estimated AMS spectra are a lot higher. A detailed discussion about the differences of the 30 min and the 24 h spectra, and a corresponding figure have been added to the paper.

**(8)** For 2- and 4-hr averaging intervals, the theta angles for all factors except BBOA are considerably small. Based on figure S8, I assume that a theta angle less than 10 degrees would more or less replicate the factor profile obtained at the 30-min resolution measurement. Hence, I am curious about why the theta angle for BBOA is much higher than others at these averaging intervals even though its % contribution (Figure 5) only changes extremely minimally. Where does this variation in BBOA factor profile come from? Some discussion would be useful for the paper.

A theta angle below 15 degrees indicates that the two factors are quite similar. In the case of the 4-h averages the theta angle for the BBOA factor (compared with the 30 min BBOA) is 14 degrees. In the 2-h PMF results the theta angle with the 30-min BBOA is equal to 19 degrees which shows that the two spectra have some significant differences mainly in the m/z values 18, 41 and 55. On the other hand, the signal at m/z values 60 and 73, which are characteristic of BBOA, were in good agreement for the two different temporal resolutions. A brief discussion has been added to the main paper and a new figure with the comparison between the 4-h and 30 min BBOA spectra has been added to the Supplementary Information.

**(9)** It is also not clear why: (i) the source apportionment for different time periods was not performed with constraining at least some primary factors, and, (ii) no assessment of uncertainty is made for solutions from differently averaged datasets through a bootstrap analysis. This is important.

We have followed the suggestion of the reviewer and repeated the analysis this time constraining the primary factors for both the 30 min and the 24 h resolution. Our conclusions were quite robust in this case too. The results of the constrained analysis at the two resolutions were quite consistent (discrepancies less than 15%) with each other. These additional results are now described in the revised paper.

A bootstrap analysis has been also performed to characterize the uncertainty of the PMF results. The estimated uncertainty of the average factor concentration was less than 30% of the mean value in all cases. The results of the bootstrap analysis have been added to the manuscript.

**(10)** The way this paper is written seems to suggest high time-resolution to be the truth and then checks for deviations from this "truth" by reducing the time-resolution. I am not sure whether this is the best approach to handling this comparison because both offline and online techniques provide useful, scientific information in their own right. Also, comparison results could change a lot when looking at offline high spectral-resolution AMS data instead of ACSM. The authors should defend why high time-resolution data should be taken as the baseline for comparisons.

This is a good point, and we agree that both the on-line and off-line techniques provide useful information. We do not consider the high temporal resolution results as the "truth" because clearly, they have their own errors characteristic of any source apportionment technique. We have added a few sentences to clarify this point both early in the manuscript and in the discussion of the results.

We also agree about the ACSM versus AMS point that the reviewer raises. The low resolution of the ACSM mass spectra used in the present work may represent a worse case scenario for the uncertainty of the off-line results. The situation may be even better for high resolution AMS spectra. This is a topic for future investigation, and we plan to address it in a forthcoming paper. A brief discussion of why high temporal resolution data are expected to lead to more accurate estimate of source contributions has been also added to the revised paper.

**Additional comments**

**(11)** lines 15-20: It is important to discuss which percentages are closer to the truth in authors' opinion.

A brief discussion of this point has been added to the paper.

**(12)** lines 75-78: It should be mentioned somewhere here that extraction efficiency in offline analysis can play a role in causing differences.

This issue is now mentioned in the revised paper together with the other experimental challenges that accompany the off-line analysis.

**(13)** lines 94-95: The authors should at least briefly describe the process of choosing the optimum Fpeak.

A brief discussion of the selection of the optimum Fpeak has been added to the manuscript.

**(14)** line 124: There is an older paper to cite for this: Identification of the mass spectra signature of organic aerosols from wood burning emissions, Alfarra et al, 2007 ES&T.

The recommended reference has been added.

**(15)** line 186: The BBOA mass spectrum for the 24-hr average looks more comparable to literature than the 30-min averages where e.g. f43 is nearly the same as f73 which is unusual.

This is a good point. The AMS and ACSM spectra for the same factor can be different because of the different fragmentation tables used in the analysis of the measurements of two instruments. These differences can be seen in several studies. This issue is now mentioned in the manuscript.

**(16)** lines 190-192: It is evident that LO- and MO-OOA will not compare well as the diurnal variability is driving the separation between the OOAs much more for the 30-min solution than for the 24-hr solution which is driven by day-to-day variability. If more seasons were used, then seasonal variability may have separated OOAs to some extent.

Use of more seasons in the analysis can introduce significant uncertainty because of the different dominant chemical processes leading to SOA production and also the chemical aging mechanisms of primary OA. A good example of this is the change in the COA spectrum that we have observed in the cold and warm periods in Greece due its rapid atmospheric processing in the summer. Also, previous work has often found that mixing results from different seasons can increase the uncertainty of the analysis. A brief discussion together with a few references analyzing this issue have been added to the paper.

**(17)** line 253: As I said in (9) above, not necessarily. The dataset in this analysis uses ACSM but AMS used now-a-days provide high spectral-resolution data with more spectral information.

We agree that the situation can be even better if AMS high mass resolution measurements are used for the off-line analysis. This topic will be addressed in a forthcoming paper. We have added the corresponding discussion in the paper.

**(18)** In section 3.2, discuss how MO-OOA and LO-OOA are separated in terms of their spectral signatures.

A discussion about the differences in the MO-OOA and LO-OOA spectra has been added.

**(19)** line 178: "underestimate".

The typo has been corrected.

**(20)** line 214: remove comma after "little"

The comma has been removed.

**(21)** line 239: add "of" after "tendency"

Done.